# HighlightRemover: Spatially Valid Pixel Learning for Image Specular Highlight Removal

## ABSTRACT

Recently, learning-based methods have made significant progress for image specular highlight removal. However, many of these approaches treat all the image pixels uniformly, overlooking the negative impact of invalid pixels on feature reconstruction. This oversight often leads to undesirable outcomes, such as color distortion or residual highlights. In this paper, we propose a novel image specular highlight removal network called HighlightRNet, which utilizes valid pixels as references to reconstruct the highlight-free image. To achieve this, we introduce a context-aware fusion block (CFBlock) that aggregates information in four directions, effectively capturing global contextual information. Additionally, we introduce a location-aware feature transformation module (LFTModule) to adaptively learn the valid pixels for feature reconstruction, thereby avoiding information errors caused by invalid pixels. With these modules, our method can produce high-quality highlight-free results without color distortion and highlight residual. Furthermore, we develop a multiple light image-capturing system to construct a large-scale highlight dataset called NSH, which exhibits minimal misalignment in image pairs and minimal brightness variation in non-highlight regions. Experimental results on various datasets demonstrate the superiority of our method over state-of-the-art methods, both qualitatively and quantitatively.

## CCS CONCEPTS

• **Computing methodologies → Computer vision problems**.

## KEYWORDS

Image specular highlight removal, contextual information, valid pixels

## 1 INTRODUCTION

Specular highlights are natural occurrences when light strikes an object with smooth surface. However, the presence of continuous or discontinuous spots in specular highlight regions often leads to poor visibility and incoherent diffuse regions in images. This phenomenon significantly increases the complexity and difficulty of various vision tasks, including object detection [13], semantic segmentation [3], object tracking [6], image segmentation [12], and so on. Therefore, effectively removing specular highlights from

*MM '24, October 28 – November 1, 2024, Melbourne, Australia*
© 2024 Copyright held by the owner/author(s). Publication rights licensed to ACM.
ACM ISBN 978-1-4503-XXXX-X/18/06...$15.00
https://doi.org/XXXXXXX.XXXXXXX

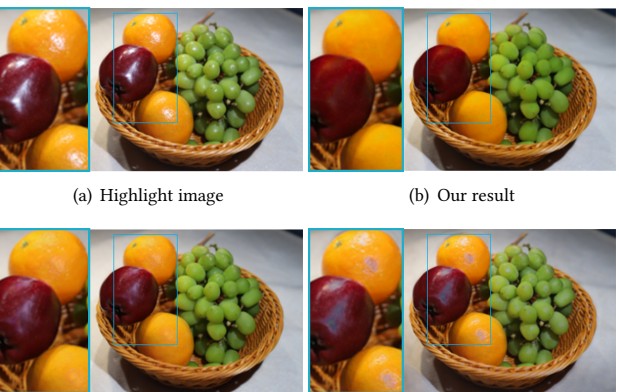

(a) Highlight image      (b) Our result

(c) Result of [27]      (d) Result of [4]

**Figure 1: Image specular highlight removal. With the consistent feature manipulations, results of [27] and [4] may cause color distortion or highlight residual. In contrast, our method can produce more desirable result by utilizing valid pixels.**

images and recovering clear, highlight-free images is both important and challenging.

Existing image specular highlight removal methods fall into two groups. Traditional methods [12, 24, 30, 31] leverage various constraints or assumptions to remove highlights from images but often demonstrate limited effectiveness. Recently, numerous learning-based specular highlight removal methods have been developed [4, 9, 10, 27]. They dig into the mapping relationship between highlight images and non-highlight images, aiming for enhanced performance. However, most of these methods uniformly process all pixels in the image, creating potential problems such as convolution of invalid pixels or deviation calculation of features. The main reason is that, the highlight regions with strong light spots are corrupted regions, and pixels in these region are invalid pixels for specular highlight removal. Simply mapping the features via consistent processing contains convolution of invalid pixels, resulting in mean and variance shifts in normalized features. It can result in invalid or biased recovery in highlight regions, leading to unsatisfactory results with highlight residual or color distortion, as shown in Figure 1(c) Figure 1(d).

Moreover, the dataset has a crucial impact on the performance of learning-based models. Currently, there are only three benchmark datasets publicly available for specular highlight removal. However, these datasets still have quality defects. For example, SHIQ [4] and SSHR [5] are synthesized datasets. But the synthetic images still exhibit some statistical feature differences from the real images. On the other hand, PSD [27] is a real-world dataset, while the image pairs in this dataset suffer from obvious misalignment and brightness variations in non-highlight regions.

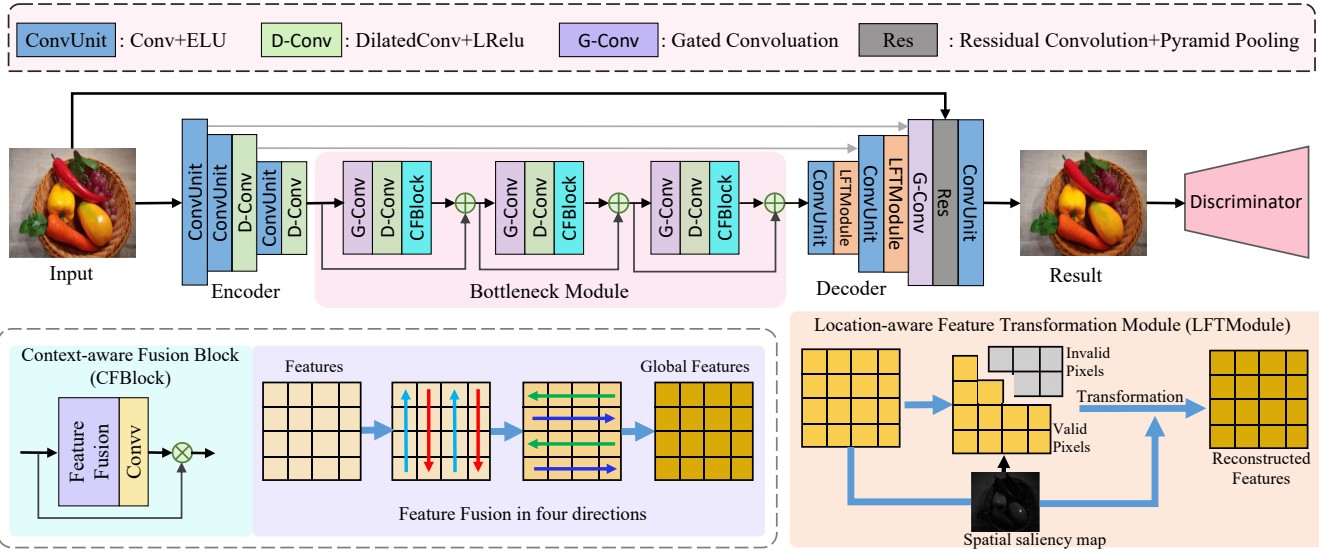

**Figure 2: The framework of the proposed HighlightRNet. We first use an encoder to extract features. Then, we introduce a context-aware fusion block (CFBlock) in the bottleneck layer to learn global contextual information. Next, we embed two location-aware feature transformation modules (LFTModule) into the decoder, aiding in the reconstruction of high-quality highlight removal results with a consistent appearance.**

To address the above challenges, we propose a novel image specular highlight removal network called HighlightRNet, which utilizes valid pixels in the image to reconstruct the highlight-free image. Figure 2 illustrates the framework of the proposed HighlightRNet, which is an encoder-decoder structure with a discriminator. Specifically, we introduce a context-aware fusion block (CFBlock) in the bottleneck module, which learns global contextual information in four directions and passes feature information from each pixel to the others. After several convolutions, the highlight region is gradually recovered, resulting in a distinct appearance from the original image. To this end, we propose a location-aware feature transformation module (LFTModule). Based on the spatial relationship of features, this module learns a spatial saliency map to demonstrate which are the valid pixels for specular highlight removal task. Thus, we can redecode the features using the valid pixels as references, avoiding information error caused by invalid pixels and promoting high-quality highlight-free results without color distortion and highlight residual, as shown in Figure 1(b).

Additionally, we construct a new large-scale real-world highlight dataset for specular highlight removal. To obtain high-quality highlight image pairs, we build a simple yet effective image-capturing system with multiple light sources. This multiple light source combination mechanism effectively avoids problems such as misalignment between image pairs and inconsistent brightness in non-highlight regions. Our image-capturing system is portable and suitable for indoor and outdoor use.

To sum up, our contributions are summarized as follows:

- We propose a network called HighlightRNet to remove specular highlights in the image, which can recover a high-quality highlight removal results without color distortion and highlight residual.

- We introduce a context-aware fusion block to learn global contextual information and a spatial feature redecoding module to reconstruct the image features using valid pixels as references.

- We construct a real-world highlight dataset without misalignment between image pairs and consistent brightness in non-highlight regions. Experimental results and evaluations demonstrate the superiority of our method over the state-of-the-art methods.

## 2 RELATED WORK

Traditional methods for image specular highlight removal often rely on additional prior knowledge [8, 17, 26]. Shafer et al. [21] introduced a method to analyze standard color image to estimate the amount of interface (specular) and body (diffuse) reflection at each pixel. Klinker et al. [15] used the difference between the object color and highlight color to separate the color of every pixel into a matte component and a highlight component. Shen et al. [22] separated reflections in a color image based on the error analysis of chromaticity and the appropriate selection of body color for each pixel. Yang et al. [29] proposed a novel reflection components separation model based on H-S color space. Yang and Tang [30] formulated the highlight removal problem as an iterative bilateral filtering process. The method proposed by Kim et al. [12] was based on an observation that the dark channel usually provides an approximate highlight-free image. Shen and Zheng [23] considered color space to analyze the distribution of the diffuse and specular components and used this information for separation. Akashi [1] proposed a model-driven approach to improve the lighting normalization of face images. Zhang et al. [32] formulated highlight detection as a Non-negative Matrix Factorization (NMF) problem.

With the development of deep learning, numerous learning-based methods have been proposed for image specular highlight removal, showing promising results using annotated training data. Lin *et al.* [16] proposed a fully-convolutional neural network (CNN), which automatically and consistently removes specular highlights from a single image by generating its diffuse component. Muhammad *et al.* [18] introduced Spec-Net, which took an intensity channel as input to remove high-intensity specularity from low chromaticity images. They also proposed Spec-CGAN, which input an RGB image to produce a diffuse image. Wu *et al.* [27] presented a novel GAN for specular highlight removal with the guidance of the detected specular reflection information. Fu *et al.* [4] developed a multi-task network for joint highlight detection and removal based on a new specular highlight image formation model. These methods can handle small sizes as well as weak highlights, but they still perform poorly for others and often carry for color or texture distortion. More recently, Fu *et al.* [5] proposed a three-stage specular highlight removal network, which first decomposed the input image into the albedo, shading, and specular residue components. Such treatment may causes the error accumulation and reduces the performance of the subsequent highlight removal due to intrinsic decomposition is also a difficult task.

## 3 NSH DATASET CONSTRUCTION

There are several image specular highlight datasets available, such as, SHIQ [4], PSD [27], and SSHR [5]. Table 1 summarizes the general information of the datasets. However, they still have some limitations:

- **SHIQ dataset:** The highlight-free images in SHIQ are computationally synthesized, with feature differences from the real-world images. In addition, this dataset lacks images with highlights caused by color illumination.
- **PSD dataset:** The variety of images is small and the background is simple. Some specular-free images have thin highlight residual. Also, the image pairs have misalignments and brightness variation in non-highlight regions.
- **SSHR dataset:** The images are rendered in software to simulate real images that have simple textures . The backgrounds in the images are blank and filled with black color, and the visual effects are lacking in realism.

In summary, the existing specular highlight datasets are still imperfect. To address this problem, we build an image-capturing system and construct a new and high-quality large-scale specular highlight dataset for image highlight removal. Our dataset is constructed on real scenes, and our image pairs have consistent brightness in non-highlight regions without misalignment.

### 3.1 Image-capturing System

The common light source in the real world is natural light, which is unpolarized light. Existing techniques [19, 27] often use cross polarizers to capture specular highlight images. In a strict laboratory environment [30], they convert a light source to linearly polarized light by adding a linear polarizer in front of the light source, as shown in Figure 3(a). When linearly polarized light strikes an object, it produces linearly polarized specular reflection and unpolarized diffuse reflection [2, 19]. As these two different reflection lights

**Table 1: Image specular highlight datasets.**

| Dataset | Amount | Content of Images | DataType |
|---------|--------|-------------------|----------|
| SHIQ | 16K | Specular/Specular-free /Specular mask | Synthetic |
| PSD | 11.7K | Specular/Specular-free | Real |
| SSHR | 130K | Specular/Specular-free /Albedo/Shading/Tone correction/Specular residue | Synthetic |
| Our NSH | 30K | Specular/Specular-free | Real |

pass through a linear polarizer, the observed image $I$ can be represented as a linear combination of a constant diffuse reflection component $I_d$ and a specular reflection component $I_s$, where $I_s$ is modulated according to the polarization of the filter [25]. Based on the dichromatic reflection model [20], the observed image $I$ can be expressed as:

$$I = \frac{1}{2}I_d + I_s \cos^2 \phi \,, \tag{1}$$

where $\phi$ is a special angle between the two polarizers, as shown in Figure 3(a).

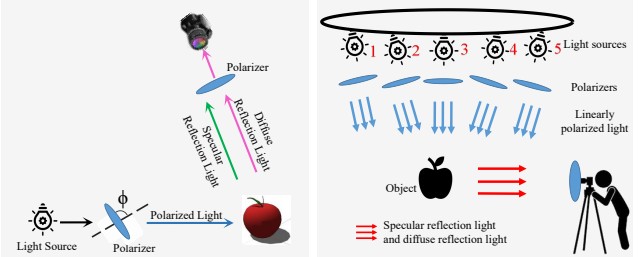

(a) Image capture with one light      (b) Image capture with multiple lights

**Figure 3: Specular highlight image captured process.** $\phi$ **in (a) is a special angle between the two polarizers.**

When capturing highlight images, we usually place a polarizer in front of both the camera and the light source. To prevent camera shake, we fix the polarizer in front of the camera, and rotate the polarizer in front of the light source to get the observed image. Wu *et al.* [27] use this strategy to construct PSD dataset. They capture the specular highlight image with $\phi = 0$ and get the corresponding diffuse image with $\phi = \frac{\pi}{2}$:

$$I = \begin{cases} \frac{1}{2}I_d + I_s, & \phi = 0 \\ \frac{1}{2}I_d, & \phi = \frac{\pi}{2} \end{cases} . \tag{2}$$

As we know, objects are basically non-Lambertian. When the linearly polarized light source strikes the object, the object's surface is usually divided into the highlight regions and the non-highlight regions. While $\phi = \frac{\pi}{2}$, linear specular reflections are filtered out, resulting in significant brightness variations in non-highlight regions for the image pairs. As shown the first heat map in Figure 5, the image pair from PSD has significant brightness variations in non-highlight regions. To solve this problem, we add the number of light sources to increase the diffuse reflection components, as

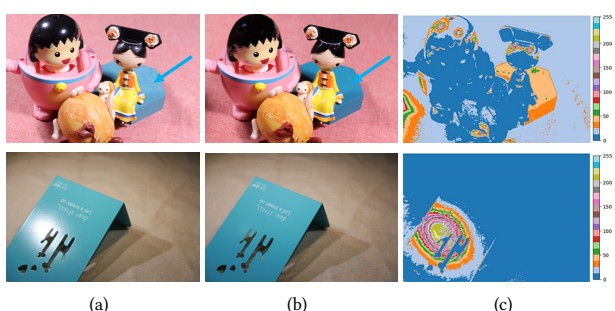

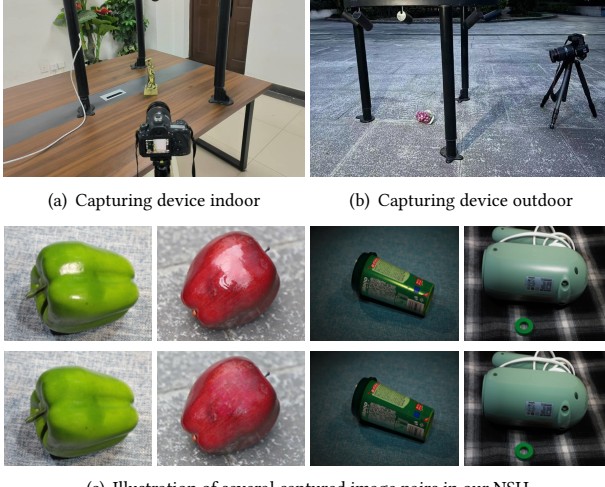

(a) Capturing device indoor  (b) Capturing device outdoor

(c) Illustration of several captured image pairs in our NSH

**Figure 4: Our image-capturing system and the captured image pairs in NSH. The top row in (c) is highlight images, and the bottom is corresponding highlight-free images (ground-truth).**

(a)  (b)  (c)

**Figure 5: Comparison between the PSD and our NSH datasets. (a) and (b) show the highlight and corresponding highlight-free images. The image pair in the first row is from PSD, and the second row is from our NSH. (c) illustrates the heat maps, highlighting the differences between the highlight and highlight-free images. The heat map from our NSH consistently displays smaller values in non-highlight regions, indicating superior image pairs.**

shown in Figure 3(b). The superposition of light reflection components is a very complex process, and here we view the process as a linear one. Assuming there are $n$ light sources in the environment, the observed image can indicate that,

$$ I = \frac{1}{2}I_{d_1} + \cdots + \frac{1}{2}I_{d_n} + I_{s_1}\cos^2\phi_1 + \cdots + I_{s_n}\cos^2\phi_n , \quad (3) $$

where $I_{d_i}$ and $I_{s_i}$ are the diffuse reflection component and the specular reflection component produced by the $i$-th light source, and $i \in \{1, \cdots, n\}$. $\phi_i$ is the special angle between the two polarizers in front of the $i$-th light source and the camera.

Assuming the $n$ light sources have the same intensity, the $n$ light sources have the same diffuse reflection component $I_d$ and specular reflection component $I_s$. Thus, Eq. 3 can be rewritten as,

$$ I = \frac{n}{2}I_d + \sum_{i=1}^{n} I_s \cos^2\phi_i . \quad (4) $$

To obtain the image pair, we set $\phi_k = 0$ for the $k$-th light source, and the special angles of the other light sources are set to $\frac{\pi}{2}$. Then, the image pair is that,

$$ I = \begin{cases} \frac{n}{2}I_d + I_s, & \phi_k = 0, \phi_j = \frac{\pi}{2}, j \in \{1, \cdots, n\} \land j \neq k \\ \frac{n}{2}I_d, & \phi_i = \frac{\pi}{2}, i \in \{1, \cdots, n\} . \end{cases} \quad (5) $$

Given sufficient light sources, the diffuse reflection components tend to infinity, and the effect of specular reflections on non-highlight regions is relatively small. At this time, if the specular reflections are filtered out, the brightness in the non-highlight regions will not change significantly. As shown the heat maps in Figure 5, our image pair remains unchanged from one another in non-highlight regions. Furthermore, we use a tripod to fix the camera and use a wireless trigger to control the captured process of the image, avoiding camera shake and misalignment in the image pair due to manual camera manipulation.

To obtain a high-quality real-world dataset for image specular highlight removal, we built a simple yet effective image-capturing system, which consists of five light sources and a Conon 6D Mark II camera in a lighting-controlled environment, as shown in Figure 4(a, b). Note that, our image-capturing system is a movable device. We can move it to the desired environment for image capture, both indoors and outdoors.

## 3.2 Dataset Collection

Our image collection process mainly includes the following four steps: 1) we place the image-capturing device in the desired environment; 2) we fix a rotatable polarizer in front of both each light source and the camera; 3) we adjust the illumination direction and place an object in the intersection area of beams; 4) the image pair is captured by controlling the location of the light source and the polarizer. Specifically, according to Eq. 5, we first set all the light source's polarizers with $\phi = \pi/2$ to obtain a diffuse image (highlight-free image). Then, we rotate the polarizer of one of the light sources with $\phi = 0$ to obtain a specular highlight image.

Repeating this process, we finally collect 30K image pairs from 3350 different scenes featuring a wide variety of materials that can easily produce highlights in daily life. Each image pair contains a highlight image and a corresponding highlight-free image. These images are divided into three parts: 22K pairs for training, 6K for testing, and 2K for validation. Figure 4(c) presents some highlight and highlight-free image pairs in our NSH.

## 4 PROPOSED METHOD

We propose an image specular highlight removal network called HighlightRNet, which leverages valid pixels in the image to reconstruct the highlight-free image. To better recognize the valid pixels, we first introduce a context-aware fusion block (CFBlock) to learn global contextual information in four different directions. Then, we propose a location-aware feature transformation module (LFTModule) to reconstruct the features using valid pixels as referents.

Our HighlightRNet is an encoder-decoder structure stacked with a discriminator, as shown in Figure 2. The encoder employs a ConvUnit and two ConvUnit+D-Conv to extract features from the image. Each ConvUnit comprises a convolution operation followed by a LRelu function, while D-Conv represents a dilated convolution with a LRelu function. The bottleneck module consists of three fusion blocks, and each fusion block applies a gated convolution and a D-Conv alongside a CFBlock. There is a residual connection between two neighboring fusion blocks. The decoder employs two ConvUnit+LFTModule layers, followed by a gated convolution, a residual module, and a ConvUnit to reconstruct the highlight-free images.

Our discriminator is a binary classifier [11] to determine whether the predicted result is real or fake. It consists of six Conv+BN+ReLu layers and a fully connected layer. The final fully connected layer employs a sigmoid function to output the actual probability of the input image.

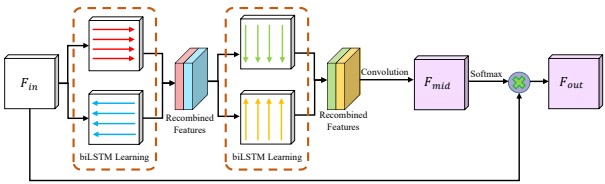

**Figure 6: The network for our context-aware fusion block (CFBlock).**

## 4.1 Context-aware Fusion Block

The convolution operations typically operate in localized regions, which can limit the extraction of global contextual features. For tasks like specular highlight removal, these localization-based convolutions may not capture contextual associations over longer distances, leading to color or texture distortion in the results. To address this issue, we introduce a context-aware fusion block (CFBlock) to learn and fuse contextual information in four different directions, enabling more effective utilization of global information.

Figure 6 illustrates the architecture of the proposed CFBlock. Initially, we segment the input features $F_{in} \in \mathbb{R}^{H \times W \times C}$ along the height dimension, where $H$, $W$ and $C$ are height, width and number of channels, respectively. We then employ bidirectional LSTM (biLSTM) [7] to learn the segmented features leftward and rightward pixel-by-pixel, enabling each pixel to retain its left and right contexts. Following biLSTM processing, we recombine the learned features. Subsequently, the combined features are split along the width dimension, and we conduct upward and downward pixel-by-pixel learning on the segmented features using biLSTM. After that, we recombine the learned features to obtain a new feature map $F_{context}$. By alternately scanning horizontally and vertically, our CFBlock effectively fuses contextual features in four directions and pass them from each pixel to the others, facilitating the perception of global contextual information.

Next, we apply a convolution to transform $F_{context}$ to $F_{mid} \in \mathbb{R}^{H \times W \times Z}$, and $Z = H \times W$. Consequently, we can obtain a feature vector $f$ for each pixel. We perform a softmax operation to normalize $f$ along the channel dimension and obtain the contextual

attention weights $\lambda$, which is that:

$$\lambda_i = \frac{exp(f_i)}{\sum\limits_{j=1}^{Z} exp(f_j)} , \qquad (6)$$

where $i \in \{1, \cdots, Z\}$, and $\lambda \in \mathbb{R}^Z$.

Finally, we perform matrix multiplication of $F_{in}$ and $\lambda$ to construct the global contextual features $F_{out}$:

$$F_{out} = \sum_{i=1}^{Z} \lambda_i F_{in} . \qquad (7)$$

## 4.2 Location-aware Feature Transformation Module

Typically, the decoder utilizes all the computed features to reconstruct the highlight-free image. However, it's crucial to note that the regions with strong specular highlights are corrupted regions, and pixels in these regions are invalid pixels for specular highlight removal. Processing all features uniformly may introduces invalid or biased convolution of pixels, potentially leading to errors in feature computation and the generation of undesirable removal results, such as color distortion and highlight residual.

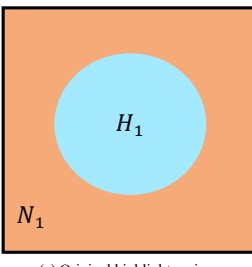 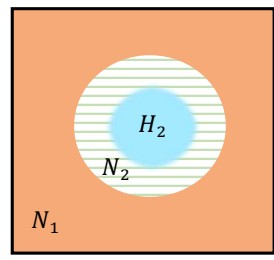

(a) Original highlight region     (b) Highlight region after several convolutions

**Figure 7: The specular highlight region. $N_1$ and $H_1$ denote the original non-highlight region and the highlight region. After passing through several convolutions, highlights in $N_2$ have been removed, and $N_2$ can be considered as a non-highlight region. $H_2$ is the remaining highlight region.**

Moreover, after several convolution operations, the specular highlights are gradually removed, as shown in Figure 7. That indicates that the specular highlight regions dynamically change during the decoding process. The repaired contents, such as $N_2$ in Figure 7(b), can also be considered as valid pixels for the restoration of the remaining highlighted regions.

Based on the preceding analysis, we introduce a location-aware feature transformation module (LFTModule), which reconstructs features using the valid pixels as references. LFTModule utilizes the spatial relationship of input features to learn a spatial saliency map, which can be considered as the distribution and the intensity of the highlights at the current layer. Larger values in the spatial saliency map indicate stronger highlights at the current position. The stronger the highlight, the higher the probability that it is an invalid pixel. With the spatial saliency map, we can recognize which are the valid pixels for feature reconstruction. Thus, we can selectively manipulate the image features using the valid pixels as

**Table 2: Quantitative comparisons of highlight removal on NSH, SHIQ and PSD datasets. ↑ means the larger the better. The best results are marked in bold.**

| Methods | Venue/Year | NSH | | SHIQ | | PSD | |
|---|---|---|---|---|---|---|---|
| | | SSIM↑ | PSNR↑ | SSIM↑ | PSNR↑ | SSIM↑ | PSNR↑ |
| Yamamoto *et al.* [28] | MTA/2019 | 0.683 | 14.651 | 0.820 | 20.513 | 0.697 | 19.897 |
| Shen *et al.* [23] | AO/2012 | 0.872 | 20.638 | 0.811 | 20.621 | 0.732 | 19.438 |
| Yang *et al.* [31] | CV/2010 | 0.633 | 19.651 | 0.776 | 17.323 | 0.753 | 15.942 |
| Wu *et al.* [27] | TMM/2021 | 0.899 | 29.921 | 0.875 | 28.637 | 0.910 | 29.153 |
| Fu *et al.* [4] | CVPR/2021 | 0.903 | 28.893 | 0.899 | 29.732 | 0.870 | 27.846 |
| Fu *et al.* [5] | ICCV/2023 | 0.901 | 26.211 | 0.917 | 27.475 | 0.897 | 26.274 |
| HighlightRNet | ACMMM/2024 | **0.942** | **30.672** | **0.930** | **30.231** | **0.922** | **29.787** |

referents, avoiding information error caused by the consistent feature process and boosting satisfactory highlight-free results without color distortion and highlight residual.

Figure 8 illustrates the pipeline of the proposed LFTModule. For the input feature $F_{de} \in \mathbb{R}^{H \times W \times C}$, we first apply max-pooling and global average pooling along the channel axis to obtain efficient feature descriptor. We integrate the results of these two pooling operations and perform a convolution operation followed by a sigmoid function to compute a spatial saliency map $A \in \mathbb{R}^{H \times W \times 1}$. We can use the spatial saliency map $A$ to get the valid pixels through a threshold $t$. If $A(h, w) < t$, we consider pixel $(h, w)$ is a valid pixel, and $(h, w)$ is an index of $(H, W)$ axis; else, we consider this pixel to be in a strong highlight region and is an invalid pixel for specular highlight removal. In our experiments, we set $t = 0.5$.

Next, we utilize the valid pixels to normalize the input features and result in feature $M_1$. Since the spatial saliency map $A$ contains global spatial information, we use convolution operation for $A$ to learn a global representation. We perform convolution operation on $A$ respectively to obtain two parameters $\gamma$ and $\beta$. We use $\gamma$ and $\beta$ as affine parameters to perform pixel-wise affine transformation on $M_1$, obtaining the reconstructed features. With the affine transformation, our LFTModule promotes consistent-looking results of the highlight regions with the surrounding environment.

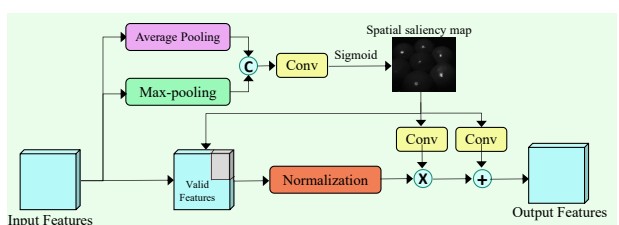

**Figure 8: The network for our location-aware feature transformation module (LFTModule).**

### 4.3 Loss Functions

The loss function for training our HightlightRNet contains three components: color consistency loss $\mathcal{L}_{color}$, texture consistency loss $\mathcal{L}_{texture}$ and adversarial loss $\mathcal{L}_{adv}$.

**Color consistency loss** is used to suppress the color distortion during the reconstruction process. It is calculated using the mean squared errors (MSE) between the predicted highlight removal result $I_{free}$ and the ground-truth image $I_{gt}$, as follows:

$$\mathcal{L}_{color} = \|I_{free} - I_{gt}\|_2^2 .$$

**Texture consistency loss** aims to preserve image structure using the gradient information in the image. It can prevent the generation of blurry results. Our texture consistency loss $\mathcal{L}_{texture}$ is calculated as,

$$\mathcal{L}_{texture} = \|\nabla_x I_{free} - \nabla_x I_{gt}\|_1 + \|\nabla_y I_{free} - \nabla_y I_{gt}\|_1 , \quad (8)$$

where $\nabla_x$ represents the gradient along the x-direction and $\nabla_y$ represents the gradient along the y-direction.

**Adversarial loss.** We employ relativistic average adversarial loss [11] to implement our adversarial loss $\mathcal{L}_{adv}$, which is described as:

$$\begin{aligned} \mathcal{L}_{adv} = 0.5 \cdot (&BCE(\sigma(D(I_{free}) - D(I_{gt})), y') \\ +&BCE(\sigma(D(I_{free}) - D(I_{gt})), y)) , \end{aligned} \quad (9)$$

where $\sigma$ is the sigmoid function and $BCE(*)$ measures the binary cross entropy. $(y', y)$ is set as $(1, 0)$ for the generator and $(0, 1)$ for the discriminator, respectively. $D$ is our discriminator.

In summary, the total loss for our method is written as:

$$\mathcal{L} = \lambda_1 \mathcal{L}_{color} + \lambda_2 \mathcal{L}_{texture} + \lambda_3 \mathcal{L}_{adv} , \quad (10)$$

where $\lambda_1$, $\lambda_2$ and $\lambda_3$ are the weighting parameters. In our experiments, we empirically set $\lambda_1 = 1.0$, $\lambda_2 = 1$, and $\lambda_3 = 0.01$.

## 5 EXPERIMENTS

### 5.1 Implementation Details

Our network is implemented in PyTorch. We use the Adam optimizer [14] to train our HightlightRNet using an NVIDIA GeForce RTX 2080 Ti GPR for 80 epochs with a batch size of 8. The initial learning rate is set to $2 \times 10^{-4}$ and is decayed by a factor of $1/2$ every 10 epochs until it reaches a value lower than $10^{-5}$.

### 5.2 Datasets and Evaluation Metrics

We evaluate the method on three datasets, including our NSH, SHIQ [4] and PSD [27]. We employ two commonly used metrics, including structural similarity index (SSIM) and peak signal-to-noise ratio (PSNR), to quantitatively evaluate the performance of our method.

### 5.3 Comparison with State-of-The-Art Methods

To verify the effectiveness of our method, we compare our method with three learning-based methods [4, 5, 27] and three traditional methods [23, 28, 31]. For a fair comparison, we directly use the codes provided by the authors with recommended parameter settings and retrain the learning-based methods on the same hardware. To train and evaluate method of Fu *et al.* [5] on the three datasets, we modify

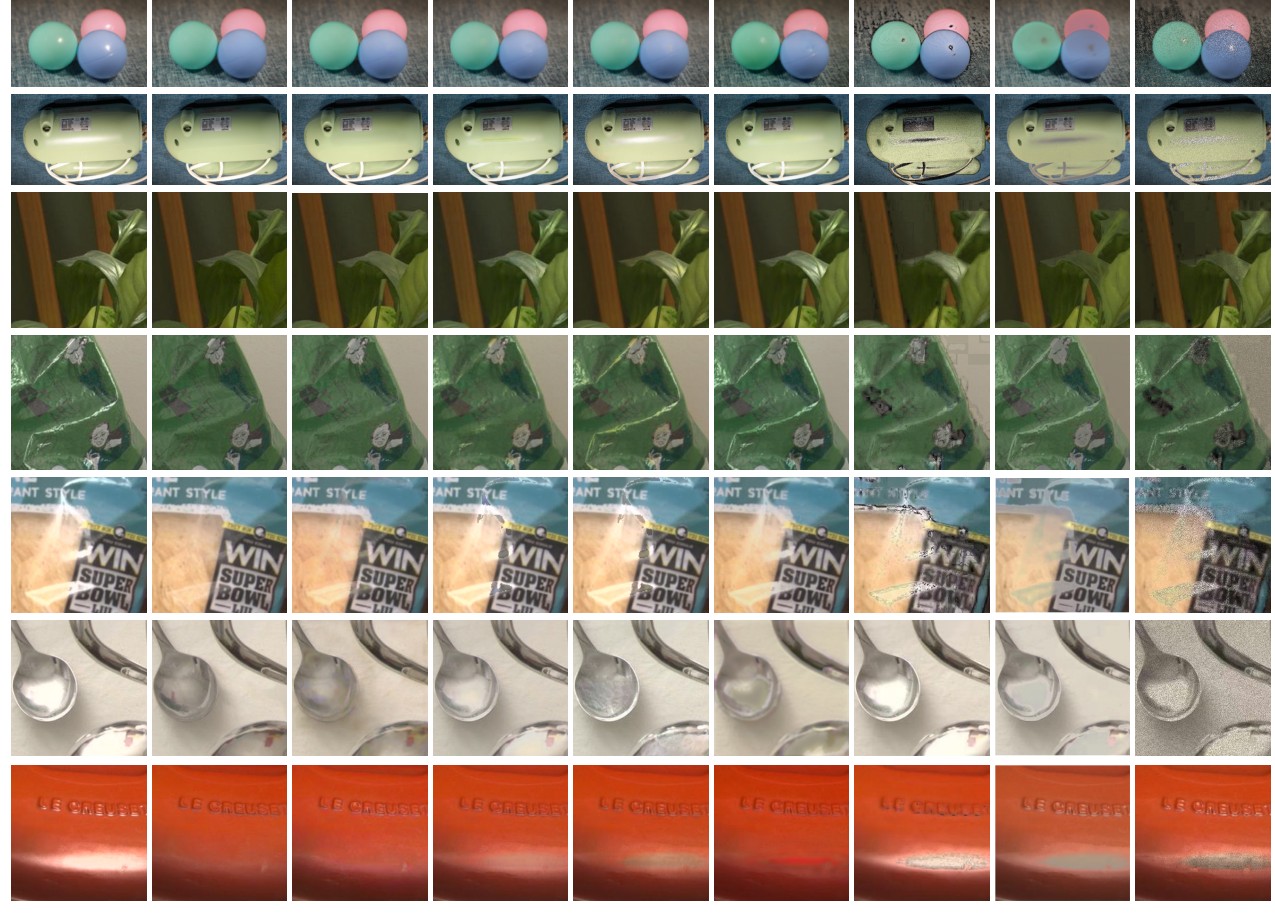

| (a) Input | (b) GT | (c) Our results | (d) Fu [5] | (e) Fu [4] | (f) Wu [27] | (g) Yang [31] | (h) Shen [23] | (i) Yamato [28] |

**Figure 9: Visual comparison of our method against state-of-the-art highlight removal methods. Compared with other results, our method produce satisfactory results without color distortion and highlight residual.**

**Table 3: Quantitative results of ablation study on NSH, SHIQ and PSD. The best results are marked in bold. ↑ means the larger the better.**

| Methods | NSH | | SHIQ | | PSD | |
|---|---|---|---|---|---|---|
| | SSIM↑ | PSNR↑ | SSIM↑ | PSNR↑ | SSIM↑ | PSNR↑ |
| $M_1$: remove CFBlock | 0.836 | 25.534 | 0.824 | 25.637 | 0.838 | 25.347 |
| $M_2$: Replace LFTModule with batchNorm | 0.901 | 26.941 | 0.877 | 26.554 | 0.873 | 25.199 |
| $M_3$: Without $\mathcal{L}_{texture}$ | 0.888 | 28.431 | 0.852 | 26.978 | 0.847 | 27.201 |
| $M_4$: Without $\mathcal{L}_{adv}$ | 0.876 | 28.433 | 0.851 | 27.207 | 0.836 | 26.954 |
| HighlightRNet | **0.942** | **30.672** | **0.930** | **30.231** | **0.922** | **29.787** |

their method and estimate the highlight-free and highlight residue instead of the original albedo and shading at the first stage.

**Quantitative Comparison.** Table 2 presents the quantitative comparisons on three datasets. From the table, we can observe that, our method achieves larger SSIM and PSNR scores on all datasets, indicating the better performance of our method compared to existing state-of-the-art methods.

**Visual Comparison.** Figure 9 illuminates some visual highlight removal results to further demonstrate the effectiveness of our method. With inaccurate shadow detection results, Fu *et al.* [4] and Wu *et al.* [27] may produce undesirable results with highlight

residual, as shown in Figure 9(e, f). Without adequate information to guide, Fu *et al.* [5] may result in color distortion or incomplete removal of highlights, as shown in Figure 9(d). Due to the lack of ability to capture high-level semantic information, the three traditional methods do not make good use of non-highlight pixels to restore the highlight regions, which usually result in color or texture distortion. For example, Yang *et al.* [30] suffer from severe artifacts such as black blocks and color distortion, as shown in Figure 9(g). Shen *et al.* [23] often result in texture loss, as shown in Figure 9(h). Yamamoto *et al.* [28] also suffer from black blocks and color distortion, as shown in Figure 9(i). Comparatively, our

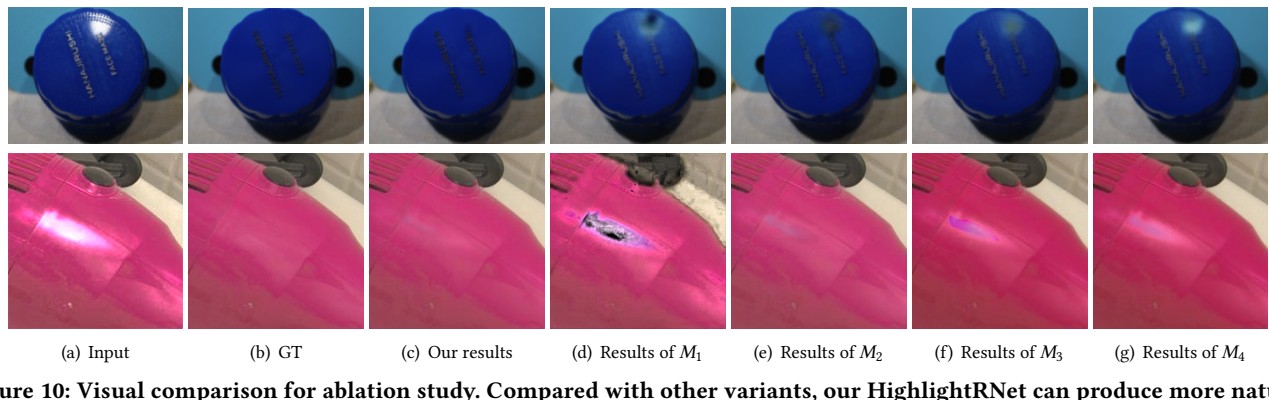

(a) Input     (b) GT     (c) Our results     (d) Results of $M_1$     (e) Results of $M_2$     (f) Results of $M_3$     (g) Results of $M_4$

**Figure 10: Visual comparison for ablation study. Compared with other variants, our HighlightRNet can produce more natural results.**

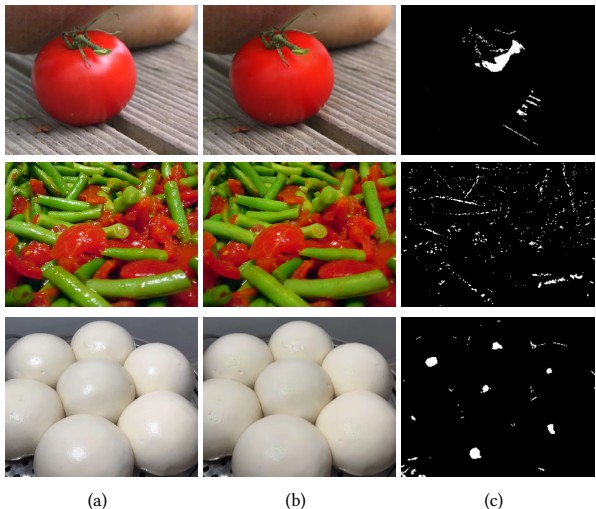

(a)       (b)       (c)

**Figure 11: Visual results for real-world natural highlight images. (a) is highlight images, and (b) is our highlight removal results. (c) is the highlight detection masks.**

method effectively removes highlight and recovers the content in the image without artifacts, which are closer to the ground truth images, as shown in Figure 9(b) and Figure 9(c).

To further verify the robustness and generalization ability of our HightlightRNet, Figure 11 presents some other highlight removal results for real-world natural images captured by smartphones or downloaded from the internet. As shown in Figure 11, our method can effectively remove the highlights and obtain natural results with few artifacts. Moreover, although we focus on highlight removal, our HighlightRNet can also be applied to detect the highlight regions based on the predicted removal result, as shown in Figure 11(c), which clearly distinguish the specular highlight regions.

## 5.4 Ablation Study

We performed a series of experiments to validate the effectiveness of our method and the superiority of our dataset.

**Effectiveness of the network.** To demonstrate the effectiveness of our HightlightRNet, we compare our network with four variants to assess the impact of each component. The variants are (1) $M_1$: remove CFBlock in HightlightRNet; (2) $M_2$: replace LFTModule

**Table 4: Ablation study about NSH dataset. The quantitative results are evaluated on NSH. The best results are marked in bold.**

| Methods | SSIM ↑ | PSNR ↑ |
|---|---|---|
| Training on SHIQ | 0.824 | 23.662 |
| Training on PSD | 0.798 | 23.512 |
| Training on SSHR | 0.844 | 25.291 |
| HighlightRNet training on NSH | **0.942** | **30.672** |

with batchNorm; (3) $M_3$: remove $\mathcal{L}_{texture}$ for training HightlightRNet; and (4) $M_4$: remove $\mathcal{L}_{adv}$ for training HightlightRNet. We train the variants on NSH. Table 3 summarizes the evaluated results on three datasets. From the table, we can observe: (1) our HightlightRNet with all components gets the best results; (2) the proposed CFBlock and LFTModule can help improve the performance of the network, and the combination leads to the best performance; and (3) the loss functions $\mathcal{L}_{texture}$ and $\mathcal{L}_{adv}$ are necessary to ensure the high-quality highlight removal results. We also provide the visualization in Figure 10, from which we can see that results produced by our HightlightRNet look more realistic with fewer artifacts.

**Superiority of NSH dataset.** To validate the superiority of our NSH dataset, we train our HighlightRNet using four datasets: SHIQ, PSD, SSHR, and our NSH datasets. Table 4 summarizes the evaluation results on NSH dataset. It is evident from the table that the model trained with our NSH dataset outperforms the models trained with the other datasets, highlighting the superiority of our NSH dataset.

## 6 CONCLUSIONS

In this paper, we propose a new network called HighlightRNet for image specular highlight removal, which utilize the valid pixels in non-highlight regions to reconstruct the highlight-free image. Particularly, we introduce a context-aware fusion block (CFBlock) to learn global contextual information in four directions. We also propose a location-aware feature transformation module (LFTModule) to adaptively learn the valid pixels for feature reconstruction, avoiding features error caused by the invalid pixels and promoting high-quality highlight-free results without color distortion and highlight residual. Experiments qualitatively and quantitatively demonstrate the superiority of our method over the state-of-the-art methods.

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
