# OpenReview forum: "HighlightRemover: Spatially Valid Pixel Learning for Image Specular Highlight Removal"
_acmmm.org/ACMMM/2024/Conference — MM2024 Poster_

### Official Review · Reviewer_bnRt · 2024-04-28

**Rating:** 5
**Confidence:** 4

**Summary:**

The paper introduces HighlightRNet, a new network designed for efficient specular highlight removal in images. HighlightRNet leverages valid pixels from non-highlight areas for superior image reconstruction. It incorporates a Context-Aware Fusion Block (CFBlock) to gather global contextual information across four directions, enhancing the quality of the reconstructed images. The network also features a Location-Aware Feature Transformation Module (LFTModule), which accurately identifies and utilizes valid pixels, minimizing common issues like color distortion and residual highlights. Additionally, the paper presents a new large-scale highlight dataset, NSH, with minimal misalignment and brightness variation, which supports the network's effectiveness. Extensive testing on various datasets shows that HighlightRNet significantly outperforms existing methods, providing high-quality, highlight-free images both qualitatively and quantitatively.

**Strengths:**

1. The author has contributed a high-quality, large-scale dataset of real-world object specular reflections, which has notably improved the effectiveness of the proposed method. This dataset is a valuable resource that will undoubtedly aid future research within this community.
2. The author introduces an innovative method and network architecture for specular reflection removal that performs exceptionally well on real-world datasets. The method has been rigorously compared with state-of-the-art techniques presented at leading conferences, showing notable improvements over previous approaches.

**Limitations:**

To ensure the comprehensive evaluation of the manuscript, I would like to inquire further regarding the methodology employed in setting the weights for the loss function. It has been noted that the author has based the weight assignment on empirical evidence. In the interest of scientific rigor, could you please clarify if there were any experiments conducted with alternative parameter settings? Additionally, it would be beneficial to have access to specific experimental results, if available, to better understand the impact of these different parameters on the overall performance of the proposed method. This information will greatly enhance the assessment of the robustness and applicability of the approach.

**Suitability:**

3

---

### Official Review · Reviewer_FprQ · 2024-05-21

**Rating:** 4
**Confidence:** 2

**Summary:**

The proposed method, HighlightRemover, introduces a network named HighlightRNet for the task of specular highlight removal in images.  This is achieved through two key innovative components: the Context-aware Fusion Block (CFBlock), designed to capture global contextual information, and the Location-aware Feature Transformation Module (LFTModule), which selectively utilizes valid pixels for feature reconstruction.  The method is underpinned by an encoder-decoder architecture with a discriminator, aimed at generating high-quality results without color distortion or highlight residual.

**Strengths:**

1. This paper develops a new dataset, NSH, which is designed to minimize misalignment and brightness variation in non-highlight regions.
2. The proposed method outperforms existing state-of-the-art techniques, both quantitatively and qualitatively comparison.
3. The authors have thoroughly evaluated their method, including comparisons with other methods and an ablation study demonstrating the contribution of different components of the proposed network.

**Limitations:**

1. While the method performs well on the NSH dataset, it is unclear how well it generalizes to other datasets or real-world scenarios that were not part of the training process.
2.  Presenting and discussing failure cases could provide a more comprehensive view of the method's robustness.

**Suitability:**

2

---

### Official Review · Reviewer_T7ws · 2024-05-23

**Rating:** 3
**Confidence:** 2

**Summary:**

This paper implements a method for high-light removal and constructs a large-scale high-light dataset (NSH). In the network structure, a CFBlock is proposed to achieve long-distance information fusion in four directions based on biLSTM, and an LFTModule is introduced to dynamically learn features for image reconstruction. Additionally, a method for collecting high-light data in real-world scenarios is also proposed.

**Strengths:**

+Significant improvements were achieved on the self-constructed dataset (NSH) as well as other datasets SHIQ and PSD;

+The paper contributed a large-scale high-light dataset and provided a detailed explanation of its shooting principles, addressing the issues of brightness consistency and alignment that appeared in previous datasets.

**Limitations:**

-Lack of innovation in network structure design:

- Using cascaded biLSTM in CFBlock to obtain long-distance information—what advantages does this have compared to the Non-local module (or transformer module)? There is a lack of relevant theoretical explanations and experimental comparisons.
- The explanation of Normalization in LFTModule is not clear. Is it a variant of BatchNormalization?
- Extracting affine parameters directly from the spatial saliency map, which contains sparse information—what is the rationale behind this design? Why is it effective?

-The proposed data construction method cannot be used for large-scale (indoor/outdoor) data preparation.

-lack comparison on the SSHR dataset and some metrics shown in the table differ significantly from those presented in other works.
[Fu G, Zhang Q, Zhu L, et al. Towards High-Quality Specular Highlight Removal by Leveraging Large-Scale Synthetic Data[C]//Proceedings of the IEEE/CVF International Conference on Computer Vision.]

-lack experiments in more complex scenarios (e.g., indoor/outdoor).

**Suitability:**

2

---

### Official Review · Reviewer_q3F1 · 2024-05-25

**Rating:** 4
**Confidence:** 3

**Summary:**

The paper introduces a unique method, HighlightRNet, which uses valid pixels for reconstructing highlight-free images. This addresses a common problem in existing methods where invalid pixels can lead to color distortion and residual highlights.

**Strengths:**

(1)The integration of the context-aware fusion block (CFBlock) and location-aware feature transformation module (LFTModule) is innovative. These components enhance the ability to capture global contextual information and adaptively learn valid pixels for feature reconstruction.
(2)The creation of the NSH dataset using a multiple light image-capturing system is a significant contribution. This dataset mitigates issues like misalignment and brightness variations found in other datasets, providing a more reliable benchmark for evaluating highlight removal methods.
(3)The paper includes extensive experimental results demonstrating the superiority of HighlightRNet over state-of-the-art methods, providing both qualitative and quantitative evidence of its effectiveness.

**Limitations:**

(1)The proposed method, while effective, may be complex to implement and require significant computational resources, which could be a barrier to practical application.
(2)The method's performance in extremely diverse and uncontrolled environments is not extensively covered. Real-world applications often involve a wide variety of lighting conditions and object textures that could affect the method’s robustness.
(3)While the NSH dataset is a valuable addition, it would be beneficial to compare its performance on more diverse datasets and under different conditions to further validate its generalizability.

**Suitability:**

2

---

### Meta-Review · Area_Chair_BpNG · 2024-07-02

**Recommendation:** Accept (Poster)
**Confidence:** 5

**Metareview:**

All reviewers give positive ratings to this work. AC decides to accept this work. The authors are suggested to consider all suggestions in their  camera ready.